# Detecting Training Data of Large Language Models via Expectation Maximization

## Abstract

The widespread deployment of large language models (LLMs) has led to impressive advancements, yet information about their training data, a critical factor in their performance, remains undisclosed. Membership inference attacks (MIAs) aim to determine whether a specific instance was part of a target model's training data. MIAs can offer insights into LLM outputs and help detect and address concerns such as data contamination and compliance with privacy and copyright standards. However, applying MIAs to LLMs presents unique challenges due to the massive scale of pre-training data and the ambiguous nature of membership. Additionally, creating appropriate benchmarks to evaluate MIA methods is not straightforward, as training and test data distributions are often unknown. In this paper, we introduce EM-MIA, a novel MIA method for LLMs that iteratively refines membership scores and prefix scores via an expectation-maximization algorithm, leveraging the duality that the estimates of these scores can be improved by each other. Membership scores and prefix scores assess how each instance is likely to be a member and discriminative as a prefix, respectively. Our method achieves state-of-the-art results on the WikiMIA dataset. To further evaluate EM-MIA, we present OLMoMIA, a benchmark built from OLMo resources, which allows us to control the difficulty of MIA tasks with varying degrees of overlap between training and test data distributions. We believe that EM-MIA serves as a robust MIA method for LLMs and that OLMoMIA provides a valuable resource for comprehensively evaluating MIA approaches, thereby driving future research in this critical area.

## 1 Introduction

Large language models (LLMs) (Brown et al., 2020; Touvron et al., 2023b) have recently emerged as a groundbreaking development and have had a transformative impact in many fields. The vast and diverse training data are central to their success, which enables LLMs to understand and generate languages to perform complex tasks. Given that training data directly shape LLMs' behaviors, knowing the composition of training data allows researchers and practitioners to assess the strengths and limitations of LLMs (Gebru et al., 2021), address ethical concerns, and mitigate potential biases (Bender et al., 2021; Feng et al., 2023) and other risks (Bommasani et al., 2021). However, the exact composition of training data is often a secret ingredient of LLMs.

Since LLMs are trained on large-scale corpora from diverse sources, including web content, the inclusion of undesirable data into training data such as test datasets (Sainz et al., 2023; Oren et al., 2023; Sainz et al., 2024), proprietary contents (Chang et al., 2023; Meeus et al., 2024c), or personally identifiable information (Mozes et al., 2023) might prevalently happen unconsciously, raising serious concerns when deploying LLMs. Membership inference attack (MIA) determines whether a particular data point has been seen during training a target model (Shokri et al., 2017). Using MIA to uncover those potential instances can serve as an effective mechanism in detecting data contamination (Magar & Schwartz, 2022) for the reliable evaluation of LLMs' ability (Zhou et al., 2023) and auditing copyright infringement (Duarte et al., 2024) and privacy leakage (Staab et al., 2023; Kandpal et al., 2023; Kim et al., 2024) to ensure compliance with regulations such as GDPR (Voigt & Von dem Bussche, 2017) and CCPA (Legislature, 2018). Therefore, MIA has gained huge interest in the LLM community.

Despite increasing demands, MIA for LLMs is challenging (Duan et al., 2024) largely due to the large scale of training data and intrinsic ambiguity of membership from the nature of languages. Furthermore, designing a proper evaluation benchmark on MIAs for LLMs that emulates a realistic test case scenario is complicated. While training data distribution is a mixture of diverse sources and unknown, target test data at inference time could be from any distribution. This circumstance motivates us to develop a robust MIA method that can work well across different conditions on the distribution of members and non-members with minimal benchmark information.

In this paper, we propose a novel MIA framework, EM-MIA, which iteratively refines membership scores and prefix scores using an expectation-maximization algorithm. A membership score indicates how likely each data point is to be a member. A prefix score indicates how discriminative each data point is in distinguishing members and non-members when used as a prefix. We empirically observe a duality between these scores, where better estimates of one can improve estimates of another. By starting from a reasonable initialization, our iterative approach progressively enhances score prediction until convergence, leading to more accurate membership scores for MIA.

To comprehensively evaluate our method and different MIA approaches, we introduce a new benchmark called OLMoMIA by utilizing OLMo (Groeneveld et al., 2024) resources. We vary degrees of distribution overlaps and control difficulty levels based on clustering. Throughout the extensive experiments, we have shown that EM-MIA is a versatile MIA method and significantly outperforms previous strong baselines, though all methods including ours still struggle to surpass random guessing in the most challenging random split setting.

Our novelty and main contributions are summarized as follows:

- To the best of our knowledge, EM-MIA is the first approach that progressively enhances membership scores (and prefix scores) with an iterative refinement for improved MIA for LLMs.
- We demonstrate that ReCaLL (Xie et al., 2024) has an over-reliance on prefix selection. In contrast, we design EM-MIA without requiring any labeled data or prior information on benchmarks.
- EM-MIA remarkably outperforms all existing strong MIA baselines for LLMs, and achieves state-of-the-art results on WikiMIA (Shi et al., 2023), the most popular benchmark for detecting pre-training data of LLMs.
- Experiments on our OLMoMIA benchmark with varying degrees of overlap between member and non-member distributions shows the robustness of EM-MIA and the utility of evaluating on diverse conditions when developing a new MIA method.

## 2 BACKGROUND

### 2.1 MEMBERSHIP INFERENCE ATTACK FOR LLMS

Membership inference attack (MIA) (Shokri et al., 2017; Carlini et al., 2022) is a binary classification task that identifies whether or not a given data point has been seen during model training: member vs non-member. Given a target language model $\mathcal{M}$ trained on an unknown $\mathcal{D}_{\text{train}}$, MIA predicts a membership label of each instance $x$ in a test dataset $\mathcal{D}_{\text{test}}$ whether $x$ in $\mathcal{D}_{\text{train}}$ or not, by computing a membership score $f(x; \mathcal{M})$ and thresholding this score. By adjusting a threshold, MIA can control the trade-off between the true positive rate and the false positive rate. The MIA performance is typically evaluated with two metrics: AUC-ROC and TPR@low FPR (Carlini et al., 2022; Mireshghallah et al., 2022).

Most existing MIA methods are based on the assumption that the target model memorizes (or overfits) training data. In general, members will have a lower loss (Yeom et al., 2018), which is the average log-likelihood (or perplexity) of target text with respect to a target LLM, compared to non-members. Likelihood Ratio Attacks (LiRAs) (Ye et al., 2022) perform difficulty calibration using a reference model (Carlini et al., 2022), a compression method (Carlini et al., 2021), or the average loss from neighbors (Mattern et al., 2023). Min-K% (Shi et al., 2023) calculates the average log-likelihood for tokens with the lowest likelihoods. Min-K%++ (Zhang et al., 2024) improves Min-K% by normalizing each token's log probability. ReCaLL (Xie et al., 2024) uses the ratio of the conditional log-likelihood to the unconditional log-likelihood as the membership score. ReCaLL is described in detail in §2.4.

## 2.2 Why MIA for LLMs is Challenging

Although MIA has been studied in machine learning for several years, it remains especially challenging for LLMs (Duan et al., 2024; Meeus et al., 2024b). Due to the massive training dataset size, each instance is used in training only a few times, often just once (Lee et al., 2021), making it difficult to leave a footprint on the model. Moreover, there is inherent ambiguity in the definition of membership because texts are often repeated and partially overlap each other in the original form or with a minor difference even after the rigorous preprocessing of decontamination and deduplication (Kandpal et al., 2022; Tirumala et al., 2024). The membership boundary becomes even fuzzier (Shilov et al., 2024) if semantically similar paraphrases (Mattern et al., 2023; Mozaffari & Marathe, 2024) beyond lexical matching based on n-gram are considered. Traditional MIA approaches in machine learning literature (Shokri et al., 2017; Ye et al., 2022; Carlini et al., 2022) based on training shadow models on non-overlapping data from the same data distribution, model architecture, and training algorithm as the target model are infeasible for LLMs considering high computational costs and unknown training specifications.

## 2.3 Evaluation of MIAs for LLMs

Common MIA benchmarks such as WikiMIA (Shi et al., 2023) use a time cutoff based on model release dates and time information of documents (Shi et al., 2023; Meeus et al., 2024a) to ensure somewhat membership labels. Besides, recent studies (Duan et al., 2024; Das et al., 2024; Meeus et al., 2024b; Maini et al., 2024) question whether several MIAs that perform well on these benchmarks are truly conducting membership inference, as detecting distributional (often temporal) shifts alone may be sufficient to achieve high benchmark performance. Thus, they advocate for the use of datasets with a random train-test split such as MIMIR (Duan et al., 2024), which is derived from the PILE dataset (Gao et al., 2020), for MIA evaluation. However, none of the existing methods significantly outperforms random guessing in this setting. Although this setting seems theoretically appropriate for evaluating MIA, there is no truly held-out in-distribution dataset in reality because LLMs are usually trained with all available data sources. In other words, it is difficult to find in-distribution non-member examples, and it is nearly impossible to completely eliminate the distribution shift between training and test data at inference time.

Once an MIA benchmark, a pair of models and datasets, is released, it becomes difficult to prevent MIAs from exploiting information about the benchmark settings, such as how the dataset has been constructed. MIAs may exploit existing biases on datasets regardless of their intended purpose. To evaluate the true generalizability of MIAs in real-world scenarios, we should consider situations where we have minimal knowledge about the target language model, training data, and test data. Several ongoing attempts (Meeus et al., 2024b; Eichler et al., 2024) aim to reproduce setups that closely resemble practical MIA scenarios. Our work is an additional effort in this direction. We propose a robust MIA method and evaluate it on our benchmark, which simulates various combinations of training and test data distribution.

## 2.4 ReCaLL: Assumptions and Limitations

ReCaLL (Xie et al., 2024) uses the ratio between the conditional log-likelihood of a target data point $x$ given a non-member prefix $p$ as a context and the unconditional log-likelihood of $x$ by an LLM $\mathcal{M}$ as a membership score, based on the observation that the distribution of ReCaLL scores for members and non-members diverges when $p$ is a non-member prefix: formally, $\mathrm{ReCaLL}_p(x; \mathcal{M}) = \frac{\mathrm{LL}(x|p;\mathcal{M})}{\mathrm{LL}(x;\mathcal{M})}$ (we may omit $\mathcal{M}$ later, for brevity), where LL is the average log-likelihood over tokens and the prefix $p$[1] is a concatenation of non-member data points $p_i$: $p = p_1 \oplus p_2 \oplus \cdots \oplus p_n$. The intuition is that the log-likelihood of members drops a lot when conditioned with non-members as a in-context learning point of view Akyürek et al. (2022), while the log-likelihood of non-members does not change much.

ReCaLL significantly outperforms other MIA approaches by a large margin. For instance, ReCaLL exceeds 90% AUC-ROC on WikiMIA (Shi et al., 2023) beating previous state-of-the-art AUC-ROC

---

[1]As an extension from using a single prefix $p$, averaging the ReCaLL scores on a set of multiple prefixes is possible for an ensemble.

of about 75% from Min-K%++ (Zhang et al., 2024). However, there is no theoretical analysis of why and when ReCaLL works well.

In Xie et al. (2024), non-members $p_i$ are randomly selected among non-members in a test dataset and excluded from the test dataset without validation, based on strong assumptions that (1) the ground truth non-members are available and (2) all of them are equally effective as a prefix. However, the held-out ground truth non-members, especially from the same distribution of the test set, may not always be available at inference time. This aligns with the discussion in the previous section §2.3. Indeed, finding non-members could be difficult (Villalobos et al., 2022; Muennighoff et al., 2024) since training data of LLMs are ever-growing by gathering all crawlable data. The solution to secure non-members in Xie et al. (2024) is generating a synthetic prefix using GPT-4o. However, according to their implementation, this method still relies on non-member test data as seed data.

Selecting non-members from the test dataset makes ReCaLL preferable and unfair compared to other MIAs that do not utilize any non-member test data. It partially explains why simple random selection reasonably works well. The ablation study from Xie et al. (2024) to demonstrate the robustness of the random prefix selection ironically reveals that ReCaLL's performance can be damaged when we do not have known non-member data points from the test dataset distribution. Using non-members from a different domain (e.g., GitHub vs. Wikipedia) significantly degrades ReCaLL's performance, sometimes even worse than Min-K%++. In other words, ReCaLL would not generalize well to test data from a distribution different from that of the prefix. The similarity between the prefix and test data also matters. Another ablation study shows a variance between different random selections, implying that random prefix selection is not consistent and that all data points are not equally effective for the prefix.

## 3 Observation: Finding a Better Prefix

In this section, we rigorously investigate how much ReCaLL's performance is sensitive to the choice of a prefix and particularly how much it can be compromised without given non-member data. We define a **prefix score** $r(p)$ as the effectiveness of $p$ as a prefix in discriminating memberships, particularly when using this prefix for ReCaLL. In the Oracle setting where ground truth labels of all examples in a test dataset $\mathcal{D}_{\text{test}}$ are available, we can calculate a prefix score by measuring the performance of ReCaLL with a prefix $p$ on a test dataset $\mathcal{D}_{\text{test}}$ using ground truth labels and a MIA evaluation metric such as AUC-ROC. While a prefix could be any text, we calculate prefix scores for all examples in a test dataset $\mathcal{D}_{\text{test}}$ by using each data point as a standalone prefix.

As an initial analysis, we conduct experiments using the WikiMIA (Shi et al., 2023) dataset with a length of 128 as a target dataset and Pythia-6.9B (Biderman et al., 2023) as a target LM. Figure 1(a) displays the distribution of prefix scores measured by AUC-ROC for members and non-members. Consistent with the results from Xie et al. (2024), ReCaLL works well if a prefix is a non-member and does not work well if a prefix is a member. Prefix scores of members are smaller than 0.7, and most of them are close to 0.5, which is the score of random guessing. Prefix scores of non-members are larger than 0.5, and most of them are larger than 0.7. This clear distinguishability suggests using a negative prefix score as a membership score.

Figure 1(b) displays ROC curves of MIA when negative prefix scores measured by different metrics are used as membership scores. We use AUC-ROC, Accuracy, and TRP@$k$%FPR with $k \in \{0.1, 1, 5, 10, 20\}$ as metrics. Using AUC-ROC-based prefix scores as membership scores achieves 98.6% AUC-ROC, which is almost perfect and the highest among other metrics.

Without access to non-members (or data points with high prefix scores), ReCaLL's performance could be significantly lower. Given the wide spectrum of prefix scores for even non-members, the effectiveness of each data point varies, and the choice of data points for a prefix can be crucial, although using a concatenation of multiple data points as a prefix reduces variance. In other words, we can expect better MIA performance by carefully selecting the prefix. Ultimately, it is desirable to find an optimal prefix $p$ without any information or access to given ground truth non-member data points on the test set.

Contrary to the Oracle setting, labels which are what should be predicted are unknown at inference time, meaning that we cannot directly use labels to calculate prefix scores. In the next section §4, we describe how our method addresses this problem by iteratively updating membership scores and

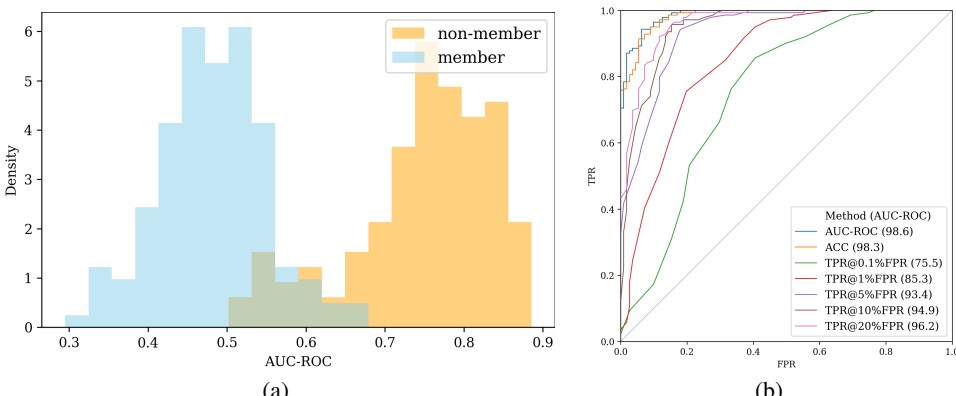

(a)                                                     (b)

Figure 1: Experiments in the Oracle setting on the WikiMIA dataset (Shi et al., 2023) with a length of 128 and Pythia-6.9B (Biderman et al., 2023) model. (a) Histogram of prefix scores for members and non-members measured by AUC-ROC. (b) ROC curves of MIA when using the negative prefix score with varying metrics as a membership score.

prefix scores based on one another. We propose a new MIA framework that is designed to work robustly on any test dataset with minimal information.

## 4 PROPOSED METHOD: EM-MIA

We target the realistic MIA scenario where test data labels are unavailable. We measure a prefix score by how $\mathrm{ReCaLL}_p$ on a test dataset $D_{test}$ aligns well with the current estimates of membership scores $f$ on $D_{test}$ denoted as $S(\mathrm{ReCaLL}_p, f, D_{test})$. More accurate membership scores can help compute more accurate prefix scores. Conversely, more accurate prefix scores can help compute more accurate membership scores. Based on this duality, we propose an iterative algorithm to refine membership scores and prefix scores via an Expectation-Maximization algorithm, called EM-MIA, to perform MIA with minimal assumptions on test data (§2.4).

Algorithm 1 summarizes the overall procedure of EM-MIA. We begin with an initial assignment of membership scores using any existing off-the-shelf MIA method such as Loss (Yeom et al., 2018) or Min-K%++ (Zhang et al., 2024) (Line 1). We calculate prefix scores $r(p)$ using membership scores and then update membership scores $f(x)$ using prefix scores. The update rule of prefix scores (Line 3) and membership scores (Line 4) is a design choice. We repeat this process iteratively until convergence (Line 5). Since EM-MIA is a general framework, all components, including initialization, score update rules, and stopping criteria, are subject to modification for further improvement.

---

**Algorithm 1** EM-MIA

**Input:** Target LLM $\mathcal{M}$, Test dataset $\mathcal{D}_{\text{test}}$
**Output:** Membership scores $f(x)$ for $x \in \mathcal{D}_{\text{test}}$
1: Initialize $f(x)$ with an existing off-the-shelf MIA method
2: **repeat**
3:   Update prefix scores $r(p) = S(\mathrm{ReCaLL}_p, f, \mathcal{D}_{\text{test}})$ for $p \in \mathcal{D}_{\text{test}}$
4:   Update membership scores $f(x) = -r(x)$ for $x \in \mathcal{D}_{\text{test}}$
5: **until** Convergence (no significant difference in $f$)

---

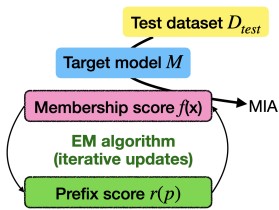

**Update Rule for Prefix Scores**   Our observation in §3 shows that AUC-ROC is an effective function $S$ to calculate prefix scores given ground truth labels. Because we do not have labels, we can assign membership labels using the current membership scores $f$ and a threshold $\tau$ to use them as approximate labels to calculate prefix scores: AUC-ROC($\{(\mathrm{ReCaLL}_p(x), \mathbf{1}_{f(x)>\tau} | x \in \mathcal{D}_{\text{test}})\}$). The value of $\tau$ could be chosen as a specific percentile in the score distributions to decide the por-

tion of members and non-members. Using a median (50% percentile) is a simple and practical choice because a test dataset is usually balanced. Instead of approximating hard labels, we can compare the ranks of $\text{ReCaLL}_p(x)$ and the ranks of $f(x)$ on $\mathcal{D}_{\text{test}}$ because the relative order among other data matters rather than the absolute values. We can use the average difference in ranks as $\sum_{x \in \mathcal{D}_{\text{test}}} \|\text{rank}(\text{ReCaLL}_p(x)) - \text{rank}(f(x))\|$ or rank correlation coefficients such as Kendall's tau (Kendall, 1938) and Spearman's rho (Spearman, 1961).

**Update Rule for Membership Scores** Our observation in §3 shows that a negative prefix score can be used as a good membership score. Alternatively, we can choose candidates with top-$k$ prefix scores to construct a prefix and calculate membership scores using ReCaLL with this prefix: $f(x) = \text{ReCaLL}_p(x)$ where $p = p_1 \oplus p_2 \oplus \cdots \oplus p_n$ and $p_i \in \text{argtopk}_{x \in \mathcal{D}_{\text{test}}} r(x)$. How to order $p_i$ in $p$ is also a design choice. Intuitively, we can place them in reverse order of the prefix score since a data point closer to the target text will have a larger impact on the likelihood.

**External data** We may extend the test dataset $\mathcal{D}_{\text{test}}$ by utilizing external data to provide additional signals. Suppose we have a dataset of known members ($\mathcal{D}_m$), a dataset of known non-members ($\mathcal{D}_{\text{nm}}$), and a dataset of instances without any membership information ($\mathcal{D}_{\text{unk}}$). For example, $\mathcal{D}_m$ could be old Wikipedia documents, sharing the common assumption that LLMs are usually trained with Wikipedia. As discussed above, we target the case of $\mathcal{D}_{\text{nm}} = \phi$, or at least $\mathcal{D}_{\text{nm}} \cap \mathcal{D}_{\text{test}} = \phi$. However, we can construct it with completely unnatural texts (e.g., "*b9qx84;5zln"). $\mathcal{D}_{\text{unk}}$ is desirably drawn from the same distribution of $\mathcal{D}_{\text{test}}$ but could be from any corpus when we do not know the test dataset distribution. Finally, we can incorporate all available data for better prediction of membership scores and prefix scores: $\mathcal{D}_{\text{test}} \leftarrow \mathcal{D}_{\text{test}} \cup \mathcal{D}_m \cup \mathcal{D}_{\text{nm}} \cup \mathcal{D}_{\text{unk}}$. We have not explored the effect of exploiting additional data in our experiments but have left it to future work.

## 5 NEW BENCHMARK: OLMoMIA

EM-MIA works well and even performs almost perfectly on some benchmarks such as WikiMIA (Shi et al., 2023) (§7.1), while it does not work well on other datasets such as MIMIR (Duan et al., 2024) similar to other methods. We want to know why this is the case and what are the conditions for success. To answer these questions, we develop a new benchmark using OLMo (Groeneveld et al., 2024), which is a series of fully open language models pre-trained with Dolma (Soldaini et al., 2024) dataset. OLMo provides intermediate model checkpoints and an index to get which data has been used for each training step, which are valuable resources to construct an MIA benchmark. We will share our implementation and resulting benchmark datasets. Like us, anyone can create their own benchmark on their purpose by modifying our implementation.

Figure 2 illustrates the basic setup of the OLMoMIA benchmark. Specifically, we use checkpoints of OLMo-7B trained with 100k, 200k, 300k, and 400k steps as target models (c.f., one epoch is slightly larger than 450k steps). Then, we consider any training data before 100k steps as members and any training data between 400k and 450k steps as non-members. However, we should keep in mind that there could still be some ambiguity in membership despite the effort of deduplication, as discussed in §2.2. Based on this setup, we have multiple settings with different samplings.

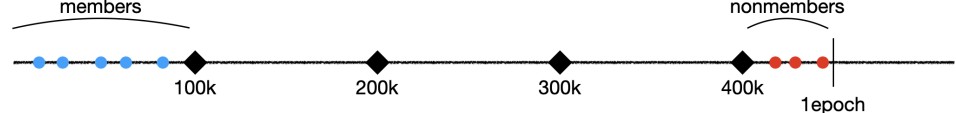

Figure 2: The basic setup of the OLMoMIA benchmark. The horizontal line indicates a training step. For any intermediate checkpoint at a specific step, we can consider training data before and after that step as members and non-members, respectively.

The most straightforward setting is *Random*, where we randomly select members and non-members from each interval. This setting is similar to MIMIR (Duan et al., 2024), where members and non-members are from the training split and test split of the Pile dataset (Gao et al., 2020), respectively. MIMIR is much more challenging than WikiMIA since the training set and test set are randomly split and thus have minimal distribution shifts.

Based on clustering, we control the degree of overlap between the distribution of members and non-members, resulting in different difficulties. First, we sample enough numbers of members and non-members (in our case, 50k each). Next, we map sampled data into embedding vectors. We use NV-Embed-v2 model (Lee et al., 2024), which was the first rank on the MTEB leaderboard (Muennighoff et al., 2022) as of Aug 30, 2024, for the embedding model. Then, we perform K-means clustering (Lloyd, 1982) for members and non-members separately (in our case, we use $K = 50$).

To prevent datasets from degeneration that have duplicates with a very minor difference, we remove data points closer than a certain distance threshold with other points within a cluster in a greedy manner to guarantee all data points in the same cluster are not too close to each other. Empirically, we observed that most pair distances measured by cosine distance range from 0.8 to 1.2, so we set 0.6 as the minimum inter-distance for deduplication.

For the *Easy* setting, we pick the farthest pair of a member cluster and a non-member cluster and pick instances farthest from the opposite cluster. For the *Hard* setting, we pick the closest pair of a member cluster and a non-member cluster and pick instances closest to the opposite cluster. For the *Medium* setting, we pick the pair of a member cluster and a non-member cluster with a median distance and randomly sampled instances within each cluster.

Additionally, we have two additional settings derived from *Random* and *Hard*: we merge members from *Random* and non-members from *Hard* to create the *Mix-1* setting, and we merge members from *Hard* and non-members from *Random* to create the *Mix-2* setting. *Mix-1* aims to simulate the case where test data come from a single cluster. Thus, any cluster might be fine, but we choose *Hard* with no specific reason. These settings cover almost all possible inclusion relationships between members and non-members.

We provide formal equations of the above descriptions in Appendix E. For each difficulty level, we have two splits based on the maximum sequence length of 64 and 128. For each dataset, we balance the number of members and non-members to 500 each to make the total size 1000.

# 6 EXPERIMENTAL SETUP

## 6.1 DATASETS AND MODELS

We evaluate EM-MIA and compare it with baselines (§6.2) on WikiMIA benchmark (Shi et al., 2023) (§7.1) and OLMoMIA benchmark (§7.2) using AUC-ROC as a main MIA evaluation metric. We also report TPR@1%FPR in Appendix B. For WikiMIA, we use the original setting with length splits of 32, 64, and 128 as test datasets and use Mamba 1.4B (Gu & Dao, 2023), Pythia 6.9B (Biderman et al., 2023), GPT-NeoX 20B (Black et al., 2022), LLaMA 13B/30B (Touvron et al., 2023a), and OPT 66B (Zhang et al., 2022) as target models, following Xie et al. (2024) and Zhang et al. (2024). For OLMoMIA (§5), we use six settings of *Easy*, *Medium*, *Hard*, *Random*, *Mix-1*, and *Mix-2* as test datasets and use checkpoints after 100k, 200k, 300k, and 400k training steps as target models. Although EM-MIA requires a baseline sufficiently better than random guessing as an initialization, there is currently no such method for MIMIR (Duan et al., 2024). Therefore, we skip experiments on MIMIR, though this is one of the widely used benchmarks on MIA for LLMs.

## 6.2 BASELINES

We compare our method against the following baselines explained in §2.1: Loss (Yeom et al., 2018), Ref (Carlini et al., 2022), Zlib (Carlini et al., 2021), Min-K% (Shi et al., 2023), and Min-K%++ (Zhang et al., 2024). We use Pythia-70m for WikiMIA and StableLM-Base-Alpha-3B-v2 model (Tow, 2023) for OLMoMIA as the reference model of the Ref method, following Shi et al. (2023) and Duan et al. (2024). For Min-K% and Min-K%++, we set $K = 20$. Among the commonly used baselines, we omit Neighbor (Mattern et al., 2023) because it is not the best in most cases though it requires LLM inference multiple times for neighborhood texts, so it is much more expensive.

### 6.2.1 ReCaLL-based Baselines

As explained in §2.4, the original ReCaLL (Xie et al., 2024) uses labeled data from the test dataset, which is unfair to compare with the above baselines and ours. More precisely, $p_i$ in the prefix $p = p_1 \oplus p_2 \oplus \cdots \oplus p_n$ are known non-members from the test set $\mathcal{D}_{\text{test}}$, and they are excluded from the test dataset for evaluation, i.e., $\mathcal{D}_{\text{test}}' = \mathcal{D}_{\text{test}} \setminus \{p_1, p_2, \cdots, p_n\}$. However, we measure the performance of ReCaLL with different prefix selection methods to understand how ReCaLL is sensitive to the prefix choice and use it as a reference instead of a direct fair comparison.

Since changing the test dataset every time for different prefixes does not make sense and makes the comparison even more complicated, we keep them in the test dataset. A language model tends to repeat, so $\text{LL}(p_i|p; \mathcal{M}) \simeq 0$. Because $\text{LL}(p_i|p; \mathcal{M}) \ll 0$, $\text{ReCaLL}_p(p_i; \mathcal{M}) \simeq 0$. It is likely to $\text{ReCaLL}_p(p_i; \mathcal{M}) \ll \text{ReCaLL}_p(x; \mathcal{M})$ for $x \in \mathcal{D}_{\text{test}} \setminus \{p_1, p_2, \cdots, p_n\}$, meaning that ReCaLL will classify $p_i$ as a non-member. The effect would be marginal if $|\mathcal{D}_{\text{test}}| \gg n$. Otherwise, we should consider this when we read numbers in the result table.

We have four options for choosing $p_i$: *Rand*, *RandM*, *RandNM*, and *TopPref*. *Rand* randomly selects any data from $\mathcal{D}_{\text{test}}$. *RandM* randomly selects member data from $\mathcal{D}_{\text{test}}$. *RandNM* randomly selects non-member data from $\mathcal{D}_{\text{test}}$. *TopPref* selects data from $\mathcal{D}_{\text{test}}$ with the highest prefix scores calculated with ground truth labels the same as §3. All options except *Rand* partially or entirely use labels in the test dataset. For all methods using a random selection (*Rand*, *RandM*, and, *RandNM*), we execute five times with different random seeds and report the average.

The original ReCaLL (Xie et al., 2024) is similar to *RandNM*, except they report the best score after trying all different $n$ values, which is again unfair. The number of shots $n$ is an important hyper-parameter determining performance. A larger $n$ generally leads to a better MIA performance but increases computational cost with a longer $p$. We fix $n = 12$ since it provides a reasonable performance while not too expensive.

Other simple baselines without using any labels are *Avg* and *AvgP*, which average ReCaLL scores over all data points in $\mathcal{D}_{\text{test}}$: $Avg(x) = \frac{1}{|\mathcal{D}_{\text{test}}|} \sum_{p \in \mathcal{D}_{\text{test}}} \text{ReCaLL}_p(x)$ and $AvgP(p) = \frac{1}{|\mathcal{D}_{\text{test}}|} \sum_{x \in \mathcal{D}_{\text{test}}} \text{ReCaLL}_p(x)$. The intuition is that averaging will smooth out ReCaLL scores with a non-discriminative prefix while keeping ReCaLL scores with a discriminative prefix without exactly knowing discriminative prefixes.

### 6.3 EM-MIA

As explained in §4, EM-MIA is a general framework where each component can be tuned for improvement, but we use the following options as defaults based on results from preliminary experiments. Overall, Min-K%++ performs best among baselines without ReCaLL-based approaches, so we use it as a default choice for initialization. Alternatively, we may use ReCaLL-based methods that do not rely on any labels like *Avg*, *AvgP*, or *Rand*. For the update rule for prefix scores, we use AUC-ROC as a default scoring function $S$. For the update rule for membership scores, we use negative prefix scores as new membership scores. For the stopping criterion, we repeat ten iterations and stop without thresholding by the score difference since we observed that membership scores and prefix scores converge quickly after a few iterations. We also observed that EM-MIA is not sensitive to the choice of the initialization method and the scoring function $S$ and converges to similar results. Ablation study on different initializations and scoring functions can be found in Appendix A.

## 7 Results and Discussion

### 7.1 WikiMIA

Table 1 and Table 3 show the experimental results on WikiMIA, with the metric of AUC-ROC and TPR@1%FPR respectively. EM-MIA achieves state-of-the-art performance on WikiMIA for all different models and length splits, significantly outperforming all baselines, including ReCaLL, even without any given non-member test data. EM-MIA exceeds 96% AUC-ROC in all cases. For the largest model OPT-66B, EM-MIA gets 99% AUC-ROC for length splits 32 and 64, while ReCaLL's performance is lower than 86% AUC-ROC.

Table 1: AUC-ROC results on WikiMIA benchmark. The second block (grey) is ReCaLL-based baselines. *RandM*, *RandNM*, ReCaLL, and *TopPref* use labels in the test dataset, so comparing them with others is unfair. We report their scores for reference. We borrow the original ReCaLL results from Xie et al. (2024) which is also unfair to be compared with ours and other baselines.

| Method | Mamba-1.4B | | | Pythia-6.9B | | | LLaMA-13B | | | NeoX-20B | | | LLaMA-30B | | | OPT-66B | | | Average | | |
|---|---|---|---|---|---|---|---|---|---|---|---|---|---|---|---|---|---|---|---|---|---|
| | 32 | 64 | 128 | 32 | 64 | 128 | 32 | 64 | 128 | 32 | 64 | 128 | 32 | 64 | 128 | 32 | 64 | 128 | 32 | 64 | 128 |
| Loss | 61.0 | 58.2 | 63.3 | 63.8 | 60.8 | 65.1 | 67.5 | 63.6 | 67.7 | 69.1 | 66.6 | 70.8 | 69.4 | 66.1 | 70.3 | 65.7 | 62.3 | 65.5 | 66.1 | 62.9 | 67.1 |
| Ref | 60.3 | 59.7 | 59.7 | 63.2 | 62.3 | 63.0 | 64.0 | 62.5 | 64.1 | 68.2 | 67.8 | 68.9 | 65.1 | 64.8 | 66.8 | 63.9 | 62.9 | 62.7 | 64.1 | 63.3 | 64.2 |
| Zlib | 61.9 | 60.4 | 65.6 | 64.3 | 62.6 | 67.6 | 67.8 | 65.3 | 69.7 | 69.3 | 68.1 | 72.4 | 69.8 | 67.4 | 71.8 | 65.8 | 63.9 | 67.4 | 66.5 | 64.6 | 69.1 |
| Min-K% | 63.3 | 61.7 | 66.7 | 66.3 | 65.0 | 69.5 | 66.8 | 66.0 | 71.5 | 72.1 | 72.1 | 75.7 | 69.3 | 68.4 | 73.7 | 67.5 | 66.5 | 70.6 | 67.5 | 66.6 | 71.3 |
| Min-K%++ | 66.4 | 67.2 | 67.7 | 70.2 | 71.8 | 69.8 | 84.4 | 84.3 | 83.8 | 75.1 | 76.4 | 75.5 | 84.3 | 84.2 | 82.8 | 69.7 | 69.8 | 71.1 | 75.0 | 75.6 | 75.1 |
| *Avg* | 70.2 | 68.3 | 65.6 | 69.3 | 68.2 | 66.7 | 77.2 | 77.3 | 74.6 | 71.4 | 72.0 | 68.7 | 79.8 | 81.0 | 79.6 | 64.6 | 65.6 | 60.0 | 72.1 | 72.1 | 69.2 |
| *AvgP* | 64.0 | 61.8 | 56.7 | 62.1 | 61.0 | 59.0 | 63.1 | 60.3 | 56.4 | 63.9 | 61.8 | 61.1 | 60.3 | 60.0 | 55.4 | 86.9 | 94.3 | 95.1 | 66.7 | 66.5 | 63.9 |
| *RandM* | 25.4 | 25.1 | 26.2 | 24.9 | 26.2 | 24.6 | 21.0 | 14.9 | 68.6 | 25.3 | 28.3 | 29.8 | 14.0 | 15.1 | 70.4 | 33.9 | 40.9 | 42.9 | 24.1 | 25.1 | 43.8 |
| *Rand* | 72.7 | 78.2 | 64.2 | 67.0 | 73.4 | 68.7 | 73.9 | 75.4 | 68.5 | 68.2 | 74.5 | 67.5 | 66.9 | 71.7 | 70.2 | 64.5 | 67.8 | 58.6 | 68.9 | 73.5 | 66.3 |
| *RandNM* | 90.7 | 90.6 | 88.4 | 87.3 | 90.0 | 88.9 | 92.1 | 93.4 | 68.8 | 85.9 | 89.9 | 86.3 | 90.6 | 92.1 | 71.8 | 78.7 | 77.6 | 67.8 | 87.5 | 88.9 | 78.7 |
| *TopPref* | 90.6 | 91.2 | 88.0 | 91.3 | 92.9 | 90.1 | 93.5 | 94.2 | 71.8 | 88.4 | 92.0 | 90.2 | 92.9 | 93.8 | 74.8 | 83.6 | 79.6 | 72.1 | 90.0 | 90.6 | 81.2 |
| Xie et al. (2024) | 90.2 | 91.4 | 91.2 | 91.6 | 93.0 | 92.6 | 92.2 | 95.2 | 92.5 | 90.5 | 93.2 | 91.7 | 90.7 | 94.9 | 91.2 | 85.1 | 79.9 | 81.0 | 90.1 | 91.3 | 90.0 |
| EM-MIA | 97.1 | 97.6 | 96.8 | 97.5 | 97.5 | 96.4 | 98.1 | 98.8 | 97.0 | 96.1 | 97.6 | 96.3 | 98.5 | 98.8 | 98.5 | 99.0 | 99.0 | 96.7 | 97.7 | 98.2 | 96.9 |

All baselines without ReCaLL-based approaches achieve lower than 76% AUC-ROC on average across different models. The relative order between ReCaLL-based baselines is consistent over different settings: $RandM < Avg, AvgP < Rand < RandNM < TopPref$. This trend clearly shows that ReCaLL is sensitive to the choice of the prefix. Particularly, the large gap between *RandM* and *Rand* versus *RandNM* shows that ReCaLL is highly dependent on the availability of given non-members. *RandNM* is similar to the original ReCaLL (Xie et al., 2024) in most cases except for the OPT-66B model and LLaMA models with sequence length 128, probably because $n = 12$ is not optimal for these cases. Among these, *Rand* does not use any labels, so it is fair to compare with other baselines, and it performs worse than Min-K%++ on average. This result again shows that ReCaLL is not strong enough without given non-members.

*TopPref* consistently outperforms *RandNM*, indicating that random prefix selection is definitely not sufficiently good and there is room for better MIA performance by prefix optimization (Shin et al., 2020; Deng et al., 2022; Guo et al., 2023). Although the search space of the prefix is exponentially large and the calculation of prefix scores without labels is nontrivial, a prefix score could be a good measure to choose data points to construct the prefix. EM-MIA approximates prefix scores and uses them to refine membership scores.

## 7.2 OLMoMIA

Table 2 and Table 4 show the experimental results on the OLMoMIA benchmark[2], with the metric of AUC-ROC and TPR@1%FPR respectively. EM-MIA achieves almost perfect scores on *Easy* and *Medium* similar to WikiMIA, gets performance comparable to random guessing performance on *Hard* and *Random* similar to MIMIR, and gets reasonably good scores on *Mix-1* and *Mix-2*, though not perfect as on *Easy* and *Medium*. EM-MIA significantly outperforms all baselines in all settings except *Hard* and *Random*, where distributions of members and non-members heavily overlap to each other and all methods are not sufficiently better than random guessing.

None of the baselines without ReCaLL-based approaches are successful in all settings, which implies that OLMoMIA is a challenging benchmark. The relative order between ReCaLL-based baselines is again consistent over different settings: $RandM < Avg, AvgP, Rand < RandNM < TopPref$, though all methods that do not use any labels fail to be successful. Interestingly, *RandNM* works reasonably well on *Mix-1* but does not work well on *Mix-2*. This is because non-members from *Mix-1* are from the same cluster while non-members from *Mix-1* are randomly sampled from the

---

[2]We expect earlier training data will be harder to detect. However, we could not find a significant difference in the MIA performance for checkpoints at different numbers of training steps. Therefore, we report average scores over four intermediate OLMo checkpoints.

Table 2: AUC-ROC results on OLMoMIA benchmark. The second block (grey) is ReCaLL-based baselines. *RandM*, *RandNM*, ReCaLL, and *TopPref* use labels in the test dataset, so comparing them with others is unfair. We report their scores for reference.

| Method | Easy | | Medium | | Hard | | Random | | Mix-1 | | Mix-2 | |
|---|---|---|---|---|---|---|---|---|---|---|---|---|
| | 64 | 128 | 64 | 128 | 64 | 128 | 64 | 128 | 64 | 128 | 64 | 128 |
| Loss | 32.5 | 63.3 | 58.9 | 49.0 | 43.3 | 51.5 | 51.2 | 52.3 | 65.7 | 49.0 | 30.8 | 54.7 |
| Ref | 56.8 | 26.8 | 61.4 | 47.2 | 49.1 | 50.7 | 49.7 | 49.9 | 59.9 | 49.7 | 38.9 | 50.9 |
| Zlib | 24.0 | 51.8 | 44.8 | 50.7 | 40.5 | 51.1 | 52.3 | 50.5 | 63.2 | 47.2 | 31.5 | 54.3 |
| Min-K% | 32.4 | 50.0 | 54.0 | 51.9 | 43.0 | 51.2 | 51.7 | 51.0 | 60.8 | 50.4 | 34.9 | 51.7 |
| Min-K%++ | 45.2 | 59.4 | 56.4 | 45.7 | 46.4 | 51.4 | 51.0 | 51.9 | 57.9 | 50.0 | 39.8 | 53.2 |
| *Avg* | 61.9 | 53.9 | 52.3 | 57.0 | 47.6 | 51.5 | 50.3 | 48.6 | 63.3 | 56.4 | 35.5 | 44.4 |
| *AvgP* | 79.2 | 39.9 | 53.9 | 61.7 | 50.2 | 51.4 | 49.0 | 50.1 | 55.7 | 63.0 | 42.7 | 41.8 |
| *RandM* | 32.3 | 22.7 | 39.2 | 30.3 | 45.8 | 50.5 | 48.1 | 48.2 | 49.7 | 48.0 | 29.1 | 28.7 |
| *Rand* | 63.7 | 46.3 | 56.0 | 59.4 | 48.9 | 52.1 | 49.7 | 49.1 | 60.6 | 68.0 | 38.0 | 38.6 |
| *RandNM* | 87.1 | 75.5 | 71.8 | 81.2 | 50.5 | 53.2 | 50.4 | 50.0 | 66.5 | 73.7 | 49.1 | 48.0 |
| *TopPref* | 88.9 | 88.5 | 79.7 | 64.4 | 55.7 | 54.5 | 52.3 | 52.7 | 79.9 | 80.2 | 55.3 | 62.1 |
| EM-MIA | 99.8 | 97.4 | 98.3 | 99.8 | 47.2 | 50.2 | 51.4 | 50.9 | 88.3 | 80.8 | 88.4 | 77.1 |

entire distribution. On the other hand, *TopPref* notably outperforms *RandNM*, implying that the effectiveness of non-member prefixes for MIA differs.

As discussed in §2.3, we reemphasize that benchmarking MIAs for LLMs is tricky because predicting where real-world test data for MIA at inference time will come from is difficult. Therefore, at least, simulating different scenarios is beneficial and necessary, like what we did using OL-MoMIA instead of a fixed existing benchmark. In this regard, we encourage MIA developers or practitioners to evaluate their methods on diverse conditions like OLMoMIA. We do not claim that OLMoMIA is closer to real scenarios than other benchmarks. However, OLMoMIA could be similar to some possible real-world scenarios even if it still does not cover all possible scenarios. The results on OLMoMIA demonstrate that EM-MIA is a robust MIA method on the varying overlap between distributions of members and non-members in a test dataset.

## 7.3 COMPUTATIONAL COSTS

MIAs for LLMs only do inference without any additional training, so they are usually not too expensive. Therefore, MIA accuracy is typically prioritized over computational costs as long as it is reasonably feasible. Nevertheless, maintaining MIAs' computational costs within a reasonable range is important. Computations on all our experiments with the used datasets (WikiMIA and OL-MoMIA) were manageable even in an academic setting. We compare computational complexity between EM-MIA and other baselines (mainly, ReCaLL) and describe how computational costs of EM-MIA can be further reduced in Appendix C.

## 8 CONCLUSION

We introduce a novel MIA method for LLMs called EM-MIA that iteratively updates membership scores and prefix scores via an Expectation-Maximization algorithm for better membership inference based on the observation of their duality. EM-MIA significantly outperforms ReCaLL even without strong assumptions that ReCaLL relies on and achieves state-of-the-art performance on WikiMIA. EM-MIA is easily tunable with several design choices, including initialization, score update rules, and stopping criteria, allowing the application to MIA in different conditions and providing room for further improvement. We create a new benchmark for detecting pre-training data of LLMs named OLMoMIA to better understand the conditions under which EM-MIA works with a comprehensive evaluation. It turns out that EM-MIA robustly performs well for all settings except when the distributions of members and non-members are almost identical, resulting in none of the existing methods being better than random guessing. We provide our thoughts on promising future research directions in Appendix D.

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

## A  ABLATION STUDY ON INITIALIZATIONS AND SCORING FUNCTIONS

Figure 3 displays the ablation study of EM-MIA with different combinations of the initialization (Loss, Ref, Zlib, Min-K%, and Min-K%++) and the scoring function $S$ () using the WikiMIA dataset with a length of 128 and Pythia-6.9B model. Each curve indicates the change of AUC-ROC calculated from the estimates of membership scores at each iteration during the expectation-maximization algorithm. In most combinations, the algorithm converges to a similar accuracy after 4-5 iterations. In this figure, there is only one case in which AUC-ROC decreases quickly and reaches a value close to 0. It is difficult to know when this happens, but it predicts members and non-members oppositely, meaning that using negative membership scores gives a good AUC-ROC.

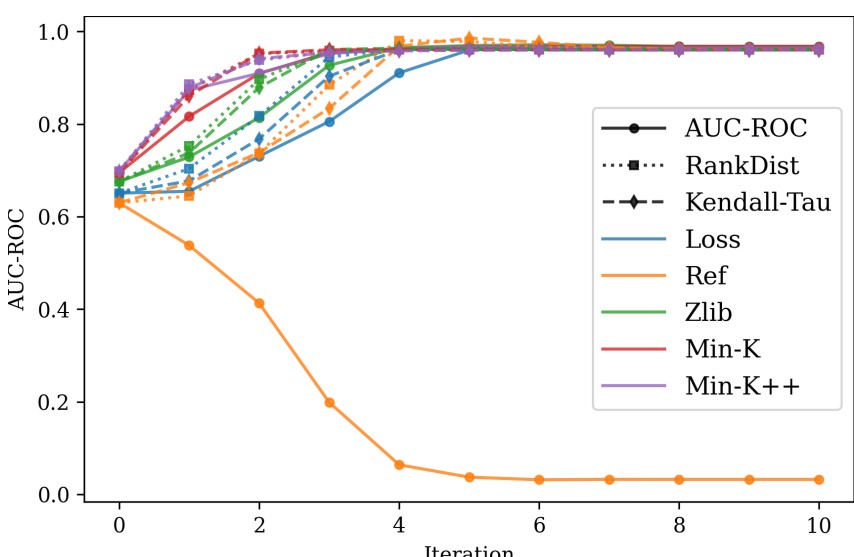

Figure 3: Performance of EM-MIA for each iteration with varying baselines for initialization and scoring functions $S$ on the WikiMIA dataset with a length of 128 and Pythia-6.9B model.

## B  TPR@1%FPR RESULTS

TPR@low FPR is a useful MIA evaluation metric (Carlini et al., 2022) in addition to AUC-ROC (§ 2.1), especially when developing a new MIA and comparing it with other MIAs. Due to the space limitation in the main text, we put TPR@low FPR here: Table 3 for WikiMIA and Table 4 for OLMoMIA.

Table 3: TPR@1%FPR results on WikiMIA benchmark. The second block (grey) is ReCaLL-based baselines. *RandM*, *RandNM*, ReCaLL, and *TopPref* use labels in the test dataset, so comparing them with others is unfair. We report their scores for reference. We borrow the original ReCaLL results from Xie et al. (2024) which is also unfair to be compared with ours and other baselines.

| Method | Mamba-1.4B | | | Pythia-6.9B | | | LLaMA-13B | | | NeoX-20B | | | LLaMA-30B | | | OPT-66B | | | Average | | |
|---|---|---|---|---|---|---|---|---|---|---|---|---|---|---|---|---|---|---|---|---|---|
| | 32 | 64 | 128 | 32 | 64 | 128 | 32 | 64 | 128 | 32 | 64 | 128 | 32 | 64 | 128 | 32 | 64 | 128 | 32 | 64 | 128 |
| Loss | 4.7 | 2.1 | 1.4 | 6.2 | 2.8 | 3.6 | 4.7 | 4.2 | 7.9 | 10.3 | 3.5 | 4.3 | 4.1 | 5.3 | 7.2 | 6.5 | 3.5 | 3.6 | 6.1 | 3.6 | 4.7 |
| Ref | 0.5 | 0.7 | 0.7 | 1.6 | 1.1 | 1.4 | 2.3 | 3.9 | 2.9 | 3.1 | 2.5 | 1.4 | 1.3 | 2.5 | 3.6 | 1.8 | 1.8 | 0.7 | 1.8 | 2.1 | 1.8 |
| Zlib | 4.1 | 4.9 | 7.2 | 4.9 | 6.0 | 11.5 | 5.7 | 8.1 | 12.9 | 9.3 | 6.3 | 5.0 | 4.9 | 9.5 | 10.1 | 5.7 | 7.0 | 11.5 | 5.8 | 7.0 | 9.7 |
| Min-K% | 7.0 | 4.2 | 5.8 | 8.8 | 3.9 | 7.2 | 5.2 | 6.0 | 15.1 | 10.6 | 3.9 | 7.2 | 4.7 | 7.0 | 5.8 | 9.0 | 7.7 | 8.6 | 7.5 | 5.5 | 8.3 |
| Min-K%++ | 4.1 | 7.0 | 1.4 | 5.9 | 10.6 | 10.1 | 10.3 | 12.0 | 25.2 | 6.2 | 9.5 | 1.4 | 8.3 | 6.7 | 9.4 | 3.6 | 12.0 | 13.7 | 6.4 | 9.6 | 10.2 |
| *Avg* | 3.9 | 0.4 | 5.0 | 8.0 | 1.1 | 7.9 | 3.1 | 7.0 | 6.5 | 6.2 | 2.1 | 8.6 | 2.8 | 6.7 | 8.6 | 2.6 | 2.1 | 4.3 | 4.4 | 3.2 | 6.8 |
| *AvgP* | 0.5 | 0.4 | 0.7 | 1.8 | 0.4 | 0.0 | 0.0 | 0.7 | 0.0 | 1.3 | 0.7 | 0.0 | 0.0 | 0.0 | 2.9 | 2.1 | 12.3 | 24.5 | 0.9 | 2.4 | 4.7 |
| *RandM* | 0.8 | 0.1 | 0.6 | 0.9 | 0.0 | 1.9 | 0.2 | 0.4 | 7.6 | 0.5 | 0.3 | 1.6 | 0.4 | 0.6 | 8.1 | 0.7 | 0.1 | 0.9 | 0.6 | 0.2 | 3.4 |
| *Rand* | 3.7 | 3.9 | 2.4 | 2.3 | 3.2 | 7.6 | 1.6 | 2.7 | 7.3 | 4.4 | 5.0 | 4.7 | 1.6 | 3.2 | 7.9 | 2.1 | 3.2 | 3.2 | 2.6 | 3.5 | 5.5 |
| *RandNM* | 19.2 | 8.3 | 15.4 | 12.6 | 10.5 | 18.7 | 18.5 | 17.2 | 7.5 | 12.9 | 11.6 | 12.5 | 13.8 | 18.7 | 8.1 | 5.0 | 5.0 | 6.6 | 13.7 | 11.9 | 11.5 |
| *TopPref* | 12.7 | 4.2 | 25.2 | 16.0 | 1.4 | 29.5 | 14.2 | 9.2 | 7.9 | 13.4 | 13.7 | 20.9 | 27.1 | 29.9 | 8.6 | 3.9 | 5.6 | 9.4 | 14.6 | 10.7 | 16.9 |
| Xie et al. (2024) | 11.2 | 11.0 | 4.0 | 28.5 | 20.7 | 33.3 | 13.3 | 30.1 | 26.3 | 25.3 | 6.9 | 30.3 | 18.4 | 18.3 | 1.0 | 8.3 | 5.3 | 6.1 | 17.5 | 15.4 | 16.9 |
| EM-MIA | 54.0 | 47.9 | 51.8 | 50.4 | 56.0 | 47.5 | 66.4 | 75.7 | 58.3 | 51.4 | 64.1 | 59.0 | 61.5 | 66.2 | 71.9 | 83.5 | 73.2 | 39.6 | 61.2 | 63.8 | 54.7 |

Table 4: AUC-ROC results on OLMoMIA benchmark. The second block (grey) is ReCaLL-based baselines. *RandM*, *RandNM*, ReCaLL, and *TopPref* use labels in the test dataset, so comparing them with others is unfair. We report their scores for reference.

| Method | Easy | | Medium | | Hard | | Random | | Mix-1 | | Mix-2 | |
|---|---|---|---|---|---|---|---|---|---|---|---|---|
| | 64 | 128 | 64 | 128 | 64 | 128 | 64 | 128 | 64 | 128 | 64 | 128 |
| Loss | 2.8 | 12.8 | 7.2 | 1.4 | 0.1 | 1.2 | 1.3 | 0.7 | 7.2 | 1.7 | 0.0 | 0.7 |
| Ref | 6.2 | 4.0 | 4.9 | 0.6 | 1.0 | 0.9 | 1.2 | 1.2 | 8.4 | 0.5 | 0.2 | 1.6 |
| Zlib | 2.0 | 9.8 | 6.7 | 1.1 | 0.2 | 1.6 | 0.9 | 0.7 | 6.4 | 1.7 | 0.0 | 0.7 |
| Min-K% | 1.3 | 6.5 | 5.8 | 1.4 | 0.1 | 1.3 | 1.1 | 0.7 | 6.1 | 2.0 | 0.0 | 0.7 |
| Min-K%++ | 1.4 | 8.0 | 5.0 | 0.7 | 0.4 | 1.0 | 1.0 | 0.4 | 5.0 | 0.9 | 0.0 | 0.5 |
| *Avg* | 4.1 | 11.5 | 4.0 | 1.7 | 0.2 | 2.2 | 1.2 | 0.6 | 6.1 | 2.2 | 0.0 | 0.9 |
| *AvgP* | 11.7 | 0.1 | 2.6 | 7.2 | 0.7 | 1.6 | 0.7 | 1.4 | 4.8 | 12.1 | 0.1 | 0.0 |
| *RandM* | 3.0 | 4.9 | 2.4 | 1.1 | 0.4 | 2.2 | 0.9 | 0.8 | 7.6 | 1.3 | 0.0 | 0.4 |
| *Rand* | 4.3 | 7.8 | 3.7 | 1.7 | 0.4 | 2.7 | 1.0 | 0.8 | 10.6 | 3.0 | 0.0 | 0.7 |
| *RandNM* | 16.9 | 14.2 | 5.2 | 1.8 | 0.3 | 1.9 | 1.0 | 0.8 | 9.2 | 2.9 | 0.0 | 1.1 |
| *TopPref* | 22.0 | 16.6 | 6.3 | 1.9 | 0.4 | 2.2 | 1.1 | 1.4 | 8.1 | 5.1 | 0.0 | 0.5 |
| EM-MIA | 95.0 | 52.1 | 79.8 | 96.7 | 1.8 | 1.0 | 1.1 | 1.4 | 12.2 | 3.8 | 14.8 | 4.3 |

## C  COMPUTATIONAL COMPLEXITY

EM-MIA is a general framework in that the update rules for prefix scores and membership scores can be designed differently (as described in §4), and they determine the trade-off between MIA accuracy and computational costs. For the design choice described in Algorithm 1 that was used in our experiments, EM-MIA requires a pairwise computation $LL_p(x)$ for all pairs $(x, p)$ once, where $x, p \in \mathcal{D}_{\text{test}}$. These values are reused to calculate the prefix scores in each iteration without recomputation. The iterative process does not require additional LLM inferences. The time complexity of EM-MIA is $O(D^2 L^2)$, where $D = |\mathcal{D}_{\text{test}}|$ and $L$ is an average token length of each data on $\mathcal{D}_{\text{test}}$, by assuming LLM inference cost is quadratic to the input sequence length due to the Transformer architecture. In this case, EM-MIA does not have other tuning hyperparameters, while Min-K% and Min-K%++ have $K$ and or ReCaLL has $n$. This is more reasonable since validation data to tune them is not given.

Of course, the baselines other than ReCaLL (Loss, Ref, Zlib, Min-K%, and Min-K%++) only compute a log-likelihood of each target text without computing a conditional log-likelihood with a prefix, so they are the most efficient: $O(DL^2)$ time complexity. Since ReCaLL uses a long prefix consisting of $n$ non-member data points, its time complexity is $O(D(nL)^2) = O(n^2 DL^2)$. According to the ReCaLL paper, they sweep $n$ from 1 to 12 to find the best $n$, which means $O((1^2 + 2^2 + \cdots + n^2)DL^2) = O(n^3 DL^2)$. Also, in some cases (Figure 3 and Table 7 in their

paper), they used $n = 28$ to achieve a better result. In theory, it may seem EM-MIA does not scale well with respect to $D$. Nevertheless, the amount of computation and time for EM-MIA with $D \sim 1000$ is not significantly larger than ReCaLL, considering the $n$ factor.

Moreover, ReCaLL requires $O(n^2)$ times larger memory than others including EM-MIA, so it may not be feasible for hardware with a small memory. In this sense, EM-MIA is more parallelizable, and we make EM-MIA faster with batching. Lastly, there is room to improve the time complexity of our method. We have not explored this yet, but for example, we may compute ReCaLL scores on a subset of the test dataset to calculate prefix scores as an approximation of our algorithm. We left improving the efficiency of EM-MIA as future work.

## D  FUTURE DIRECTIONS

While our paper focuses on detecting pre-training data with the gray-box access of LLMs where computing the probability of a text from output logits is possible, many proprietary LLMs are usually further fine-tuned (Ouyang et al., 2022; Chung et al., 2024), and they only provide generation outputs, which is the black-box setting. We left the extension of our approach to MIAs for fine-tuned LLMs (Song & Shmatikov, 2019; Jagannatha et al., 2021; Mahloujifar et al., 2021; Shejwalkar et al., 2021; Mireshghallah et al., 2022; Tu et al., 2024; Feng et al., 2024) or for LLMs with black-box access (Dong et al., 2024; Zhou et al., 2024; Kaneko et al., 2024) as future work.

## E  FORMULATION OF OLMoMIA SETTINGS

After the filtering of removing close points, let member clusters as $C_i^m$ for $i \in [1, K]$ and non-member clusters as $C_j^{nm}$ for $j \in [1, K]$. These clusters satisfy $d(x, y) > 0.6$ for all $x, y \in C_i^m$ and $d(x, y) > 0.6$ for all $x, y \in C_j^{nm}$. The following equations formalize how we construct different settings of OLMoMIA:

- *Random*: $\mathcal{D}_{\text{random}} = \mathcal{D}_{\text{random}}^{\text{m}} \cup \mathcal{D}_{\text{random}}^{\text{nm}}$
- *Easy*: $\mathcal{D}_{\text{easy}} = \mathcal{D}_{\text{easy}}^{\text{m}} \cup \mathcal{D}_{\text{easy}}^{\text{nm}}$, where $i_{easy}, j_{easy} = \arg\max_{(i,j)} \mathbb{E}_{x \in C_i, y \in C_j} d(x, y)$, $\mathcal{D}_{\text{easy}}^{\text{m}} = \arg\text{topk}_x \mathbb{E}_{y \in C_{j_{easy}}^{nm}} d(x, y)$, and $\mathcal{D}_{\text{easy}}^{\text{nm}} = \arg\text{topk}_y \mathbb{E}_{x \in C_{i_{easy}}^m} d(x, y)$
- *Hard*: $\mathcal{D}_{\text{hard}} = \mathcal{D}_{\text{hard}}^{\text{m}} \cup \mathcal{D}_{\text{hard}}^{\text{nm}}$, where $i_{hard}, j_{hard} = \arg\min_{(i,j)} \mathbb{E}_{x \in C_i, y \in C_j} d(x, y)$, $\mathcal{D}_{\text{hard}}^{\text{m}} = \arg\text{topk}_x -\mathbb{E}_{y \in C_{j_{hard}}^{nm}} d(x, y)$, and $\mathcal{D}_{\text{hard}}^{\text{nm}} = \arg\text{topk}_y -\mathbb{E}_{x \in C_{i_{hard}}^m} d(x, y)$
- *Medium*: $\mathcal{D}_{\text{medium}} = \mathcal{D}_{\text{medium}}^{\text{m}} \cup \mathcal{D}_{\text{medium}}^{\text{nm}}$, where $i_{medium}, j_{medium} = \text{median}_{(i,j)} \mathbb{E}_{x \in C_i, y \in C_j} d(x, y)$, $\mathcal{D}_{\text{medium}}^{\text{m}} \subset C_{i_{medium}}^m$, and $\mathcal{D}_{\text{medium}}^{\text{nm}} \subset C_{j_{medium}}^{nm}$
- *Mix-1*: $\mathcal{D}_{\text{mix}-1} = \mathcal{D}_{\text{random}}^{\text{m}} \cup \mathcal{D}_{\text{hard}}^{\text{nm}}$
- *Mix-2*: $\mathcal{D}_{\text{mix}-2} = \mathcal{D}_{\text{hard}}^{\text{m}} \cup \mathcal{D}_{\text{random}}^{\text{nm}}$

