# OpenReview forum: "Detecting Training Data of Large Language Models via Expectation Maximization"
_ICLR.cc/2025/Conference — Submitted to ICLR 2025_

### Official Review · Reviewer_zPaX · 2024-10-25

**Soundness:** 3
**Presentation:** 3
**Contribution:** 3
**Rating:** 6
**Confidence:** 4

**Summary:**

This paper presents a new method to improve membership inference attacks (MIAs) on large language models (LLMs) named EM-MIA.
EM-MIA is an iterative algorithm based on the expectation-maximization (EM) framework, which jointly refines membership and prefix scores to improve accuracy.
The paper also introduces OLMoMIA, a benchmark allowing controlled experiments on MIA difficulty levels by adjusting training and test data distribution overlap.
Through extensive experiments, EM-MIA achieves state-of-the-art performance on the WikiMIA dataset and outperforms existing methods (e.g., loss-based, min-k%, zlib, and ReCaLL) across various conditions in OLMoMIA.

**Strengths:**

S1. The proposed EM-MIA framework introduces a novel approach to MIAs on LLMs by leveraging the expectation-maximization algorithm to iteratively enhance membership and prefix scores.

S2. There is a comprehensive set of experiments/evaluations that compare EM-MIA to strong baselines, including ReCaLL, across multiple benchmarks such as WikiMIA and OLMoMIA.

S3. Given the growing deployment of LLMs and the increased need for privacy compliance, this work addresses a highly relevant issue in model auditing and data privacy. The authors also considered recent works that criticize MIAs on LLMs, thus also considering random splits for train/test sets (where the result is similar to other methods ~50%).

**Weaknesses:**

Naturally, the iterative nature of EM-MIA may introduce additional computational costs compared to some baselines, especially for larger datasets or LLMs. The paper could provide an analysis of the computational complexity, with timing comparisons to baselines like ReCaLL. Highlighting any trade-offs between accuracy and computational demands would help readers assess EM-MIA's scalability and practical feasibility.

**Questions:**

- Can the authors please elaborate on the computational requirements of EM-MIA relative to baselines like ReCaLL?

- The paper mentions that EM-MIA’s iterative process can be initialized with different methods, yet Min-K%++ was chosen for initialization. Could the authors please provide an ablation study or justification for this choice, and discuss how EM-MIA performs when initialized with other baselines (e.g., Loss or Avg)? This information could illustrate the robustness of EM-MIA’s initialization and its dependence on a well-performing baseline.

---

> ### Author Response · Authors · 2024-11-26
> **Rebuttal to Reviewer zPaX**
>
> We sincerely thank you for your thoughtful summary of our paper and for recognizing its strengths, particularly the novelty and key contributions in the context of recent research on MIAs for LLMs. We also greatly appreciate your constructive feedback and the opportunity to address your concerns. If our responses sufficiently clarify these points, we kindly hope you will consider raising your score.
>
> ---
>
> ### **[Weakness: Computational Costs]**
> Please see General Response (and Section 7.3 and Appendix C).
>
> ---
>
> ### **[Question: Initialization with Different Methods]**
> Please see General Response (and Appendix A).

---

> > ### Author Response · Authors · 2024-12-03
> >
> > We addressed most of your concerns and questions in our General Response.
> > We are curious whether you’ve had a chance to review it, as we haven’t received any follow-up from you since our rebuttal.
> >
> > If our response has addressed your major concerns, we kindly ask you to consider raising your score, even though your current score is already in the acceptance range.

---

### Official Review · Reviewer_HkGC · 2024-11-01

**Soundness:** 1
**Presentation:** 2
**Contribution:** 1
**Rating:** 1
**Confidence:** 5

**Summary:**

This paper tackles the problem of resolving if a given data point was used to train a large language model (LLM). This work follows a long line of previous papers that tried to solve the problem, however, as we already clearly know, it is impossible to solve it based on the result of Maini et al. ICLR 2021 [1] (see Theorem 2). The authors state at the end of their introduction that also their method is also unsuccessful in detecting members vs non-members: “Throughout the extensive experiments, we have shown that EM-MIA is a versatile MIA method and significantly outperforms previous strong baselines, though **all methods including ours still struggle to surpass random guessing in the most challenging random split setting.**” This is not a surprise based on the aforementioned Theorem 2.

**References:**

1.	Dataset Inference: Ownership Resolution in Machine Learning. Pratyush Maini, Mohammad Yaghini, Nicolas Papernot, ICLR 2021 (Spotlight).

**Strengths:**

1.	The benchmark to assess how good the MIA methods are in distinguishing between members and non-members. However, based on the aforementioned Theorem 2, all these methods fail while we train the LLMs on more data and a given sample is seen only once during the training process.

**Weaknesses:**

1.	The paper considers a simple scenario where “blind baselines can beat their membership inference attack” [1].
2.	The paper also fails on the benchmarks proposed by Maini et al. 2024 based on the training and validation splits from the Pile dataset [2].
3.	The membership inference attack is clearly defined, e.g. [3] and the assumptions made in this paper violates this basic requirement, based on the following claim: “Although this setting seems theoretically appropriate for evaluating MIA, there is no truly held-out in-distribution dataset in reality because LLMs are usually trained with all available data sources.” Thus, either another attack is proposed or the proposed attack has a random performance when claimed to be the membership inference attack.
4.	This method does not work well on the standard benchmark for membership inference on LLMs MIMIR by Duan et al 2024 (as stated at the beginning of Section 5).
5.	The designed OLMoMIA benchmark (as described in Section 5) is the same as proposed in Duan et al. 2024 as well as in Maini et al. 2024 [2]. The experiments are lacking the assessment on the Pile dataset: Section 6.1: “we skip experiments on MIMIR, though this is one of the widely used benchmarks on MIA for LLMs”
6.	No source code is provided!
7.	The results in Figure 1b with TPR@5%FPR = 93.4 are worse than “Blind Baselines” (Table 1) which report 94.4%.

**References:**

1.	Blind Baselines Beat Membership Inference Attacks for Foundation Models. Debeshee Das, Jie Zhang, Florian Tramèr https://arxiv.org/abs/2406.16201
2.	LLM Dataset Inference: Did you train on my dataset? Pratyush Maini, Hengrui Jia, Nicolas Papernot, Adam Dziedzic. NeurIPS 2024 https://arxiv.org/abs/2406.06443
3.	Membership Inference Attacks From First Principles. Nicholas Carlini, Steve Chien, Milad Nasr, Shuang Song, Andreas Terzis, Florian Tramer. S&P 2022. https://www.computer.org/csdl/proceedings-article/sp/2022/131600b519/1FlQBPf7ixy

**Questions:**

1.	Section 2.3: What is expected from the recently published papers? What kind of adoption is required? Based on this statement: “Several ongoing attempts (Meeus et al., 2024b; Eichler et al., 2024) aim to reproduce setups that closely resemble practical MIA scenarios, but none are sufficiently effective to gain widespread adoption in the community.” These papers were published this year.
2.	Section 3: “Without access to non-members (or data points with high prefix scores), ReCaLL’s performance could be significantly lower.” Would you please measure it precisely?
3.	Section 3: “We propose a new MIA framework that is designed to work robustly on any test dataset with minimal information” What is defined by working robustly? What is the minimal information?
4.	Section 4: “We target the realistic MIA scenario where test data labels are unavailable.” What are the test data labels?
5.	Section 4: “We measure a prefix score by how ReCaLLp on a test dataset D_test aligns well with the current estimates of membership scores f on D_test denoted as S(ReCaLL_p, f, D_test).” What is the D_test? How is S computed? D_test looks like the dataset for which we want to infer the membership. This should be clearly stated!
6.	What is the $\delta$ at the end of Section 5?
7.	The proposed method states that we could use something instead of clearly indicating what is used. For example, is the Kendall’s tau or Spearman’s rho used?
8.	The subsection about external data is totally not fitting to this paper. Why do you consider access to members and non-members? You make this assumption but then state that you never consider this in your experiments. If so, this subsection should be removed.

---

> ### Author Response · Authors · 2024-11-26
> **Rebuttal to Reviewer HkGC**
>
> Thank you for your detailed comments. Unfortunately, we believe some of your points stem from misunderstandings, which we address below. We also plan to clarify these aspects in the revised version of our paper.
>
> ---
>
> ### **Weaknesses**
> - W1. While “Blind Baselines” are supervised classifiers requiring labeled data (i.e., members and non-members), our EM-MIA is entirely unsupervised and does not rely on any labeled data.
> - W2. We did not evaluate on benchmarks proposed by Miani et al. (2024) because their work focuses on dataset-level inference, whereas our paper addresses instance-level membership inference, which is a different problem.
> - W3. Could you clarify your perspective? Do you believe our method qualifies as MIA or not? Furthermore, do you think it lacks practical utility?
> - W4 & W5. As noted in Duan et al. (2024), none of the existing MIAs significantly outperform random guessing. Our method requires baseline methods that perform meaningfully better than random guessing, and we have explained this requirement in our work.
> - W5. While the random setting of OLMoMIA is similar to MIMIR, OLMoMIA also includes additional settings that differ significantly from MIMIR.
> - W6. We uploaded our code as supplementary material.
> - W7. It seems there was a misunderstanding regarding Figure 1(b). This figure indicates 93.4% AUC-ROC in MIA when prefix scores are calculated using TPR@5%FPR with ground truth labels and negative prefix scores are used as membership scores. Comparison with 94.4% TPR@5%FPR of “Blind Baselines” does not make sense at all.
>
> ---
>
> ### **Questions**
> - Q1. We have removed the phrase “none are sufficiently effective to gain widespread adoption in the community.” We hope to see this line of work further.
> - Q2. We do provide precise measures of ReCaLL’s performance without access to non-members (or data points with high prefix scores). Please refer to the results of ReCaLL-based baselines (Section 6.2.1), especially the “Rand” method in our main experiment tables.
> - Q3. The robustness mainly means the method's ability to perform well across different data distributions. The minimal information includes no access to known members or non-members.
> - Q4. This is related to Q3. The test data labels mean membership labels (i.e., whether a point is a member or non-member).
> - Q5. Yes, as you understood, $D_{test}$ is the test dataset for which we aim to infer membership. The computation of the scoring function $S$ is explained in the "Update Rule for Prefix Scores" paragraph.
> - Q6. We meant an indicator function. This has been fixed in the revised manuscript.
> - Q7. We used the AUC-ROC-based scoring function $S$, as described in the formula you referred to in Q6. This is explicitly mentioned in the experimental section (Section 6.3).
> - Q8. While we did not include experiments with external data, we believe our discussion can provide avenues for future exploration.

---

> ### Comment · Reviewer_HkGC · 2024-11-26
> **Thank you for your response!**
>
> I really appreciate the effort. However, there is a clear misunderstanding of the previous work. For example:
>
> >**W2. We did not evaluate on benchmarks proposed by Miani et al. (2024) because their work focuses on dataset-level inference, whereas our paper addresses instance-level membership inference, which is a different problem.**
>
> The work by Maini et al. [1] has two main contributions. First, they show that **MIAs for LLMs do not work based on Maini et al. ICLR 2021 (SPOTLIGT) [2] (see Theorem 2)**. Second, as an alternative to membership inference, they advocated for a shift in focus toward dataset inference.
>
> Nit: It is not Miani but Maini.
>
> As explained in the initial review, this paper fails on the benchmarks proposed by Maini et al. 2024 based on the training and validation splits from the Pile dataset. Thus, I keep my score unchanged.
>
> **References:**
>
> 1. LLM Dataset Inference: Did you train on my dataset? Pratyush Maini, Hengrui Jia, Nicolas Papernot, Adam Dziedzic. NeurIPS 2024.
>
> 2.  Dataset Inference: Ownership Resolution in Machine Learning. Pratyush Maini, Mohammad Yaghini, Nicolas Papernot. ICLR 2021 (Spotlight).

---

### Official Review · Reviewer_dgtr · 2024-11-04

**Soundness:** 3
**Presentation:** 3
**Contribution:** 2
**Rating:** 3
**Confidence:** 5

**Summary:**

This paper introduces a new MIA method called EA-MIA, based on RECALL, which iteratively improves MIA scores and prefix scores via an expectation-maximization algorithm. The authors also introduce a new dataset, OLMoMIA, derived from the OLMo dataset, featuring different levels of membership inference difficulty.

**Strengths:**

- The authors employed interesting clustering techniques and embedding models to design different levels of membership inference difficulty for OLMoMIA.
- The authors provide analysis over ReCaLL assumptions and weaknesses.

**Weaknesses:**

- The authors used the WikiMIA dataset in different parts of the paper (for example, for observing which update rule to use for EA-MIA). However, as they mentioned in Section 2.3, wikiMIA is not a reliable dataset to be used for MIA experiments.

- The authors did not provide the results of their approach on the MIMIR dataset, a very well-known dataset in MIA literature. They provided this reason in Section 6.1: "Although EM-MIA requires a baseline sufficiently better than random guessing as an initialization, there is currently no such method for MIMIR (Duan et al., 2024). Therefore, we skip experiments on MIMIR, though this is one of the widely used benchmarks on MIA for LLMs". However, mink++ paper reported AUC-ROC scores of 61.1 and 74.2 for Pythia-12b and on MIMIR wikipedia and Github splits respectively, which could be used as good initializations for EM-MIA.

- Reporting TPR for low FPR is an important experiment results for a new MIA to be compared to other MIAs. This paper does not provide any results on TPR for low FPRs.

**Questions:**

- Section 3.2 contains experiments for MIA against LLaMA and OPT models using the wikiMIA dataset. I understand that these target models have not seen the non-members because of the release date. My question is: How are we sure that the members are included in their training datasets? For example, some Wikipedia articles (published before release dates) might be part of their test partition or validation partition.

- The authors mention that the concentration of non-members sometimes produces better prefix scores. What is the impact of the prefix size on prefix scores? It would be great to see an ablation study showing how different prefix sizes impact the prefix scores.

- What is the computation cost of the iterative approach of maximization. I mean in Alg 1, for each p in D_test and x in D_test, we are doing multiple round of refining r(p) and f(x). Is it expensive to these operations in multiple rounds?

- What is the impact of number of clusters in difficulty level of OLMoMIA? Why did the authors use k=50? it would be interesting to see more about the clustering hyperparameters impacts on the difficulty metric of membership inference?

- Why mink and mink++ in table 2 do not get better AUC-ROC when we are switching the difficulty level? And why they don't get better for larger target text (128 compared to 64)?

- The authors in section 6.3 mentioned that: " We also observed that EM-MIA is not sensitive to the choice of the initialization method and the scoring function S and converges to similar results" Could you please elaborate more on the intuition behind this?

- It would be interesting to see n-gram overlap ratio between members and non-members (similar to ref AA) as a difficulty metric in easy, medium and hard splits of OLMoMIA.

---

> ### Author Response · Authors · 2024-11-26
> **Rebuttal to Reviewer dgtr**
>
> We are thrilled that you appreciate the OLMoMIA dataset and recognize our improvements over ReCALL, particularly in overcoming its assumptions and limitations. We hope you will also acknowledge the significance of our main contributions, including the technical novelty of EM-MIA, in advancing research on MIAs for large language models (LLMs). If our responses address your concerns, we kindly ask you to consider raising your score. Below, we address your comments in detail.
>
> ---
>
> ### **[WikiMIA]**
> We did not explicitly mention in Section 2.3 that WikiMIA is not a reliable dataset for MIA experiments, though we noted that several papers are arguing that. Still, we believe that WikiMIA remains a valuable resource for MIA research. The prior SoTA MIAs, including Min-K%++ and ReCaLL, were developed using this dataset. Additionally, we assume that the likelihood of member data from WikiMIA belonging to the validation set is very low, as the validation set would be a tiny portion of the entire data.
>
> ---
>
> ### **[TPR @ low FPR]**
> Please see General Response (and Appendix B).
>
> ---
>
> ### **[Computational Cost]**
> Please see General Response (and Section 7.3 and Appendix C). After computing $ReCaLL_p(x)$ for all $p \in D_{test}$ and $x \in D_{test}$, these scores are reused at each iteration to refine $r(p)$ and $f(x)$. This reuse eliminates the need for additional LLM inferences, making the process computationally efficient.
>
> ---
>
> ### **[Robustness to the choice of the initialization method and the scoring function]**
> Please see General Response (and Appendix A).
>
> ---
>
> ### **[OLMo-MIA - number of clusters, Min-K%/Min-K%++, and n-gram overlap ratio]**
> We determined the number of clusters using the Silhouette score, a commonly used metric for K-means clustering.
> While Min-K% and Min-K%++ often outperform other methods, this is not guaranteed. These methods may be sensitive to the choice of $K$. In our experiments, we used a fixed value of $K=20$ without further tuning.
> We will release the OLMoMIA dataset so that researchers can use and analyze it with various metrics, including the n-gram overlap ratio. Additionally, our code will enable users to generate their own OLMoMIA datasets for further exploration.
>
> ---
>
> ### **[Prefix Sizes]**
> Could you clarify what you mean by "prefix size" and "concentration of non-members"? These terms are not used in our paper, and additional context would help us address your concern more effectively.

---

> > ### Comment · Reviewer_dgtr · 2024-11-26
> > **Official Comment by Reviewer dgtr**
> >
> > I thank the authors for their response and updates.
> >
> > sorry for misspelling, by "concentration of non-members", I meant concatenation of non-member samples (mentioned in section 3). I wanted to know what the impact of the prefix size on prefix scores when we are concatenating different number of non-members. by prefix size, I meant the number of tokens or number of non-members that are used as the prefix in this scenario.
> >
> > While I appreciate the authors' efforts, my largest weaknesses regarding wikiMIA dataset (for example, for observing which update rule to use for EA-MIA) and novelty of the paper still persists. I hence keep my score.

---

> > > ### Author Response · Authors · 2024-12-03
> > >
> > > Thank you for your reply! We are glad you appreciated our response and updates.
> > >
> > > In Section 3, we noted that each data point in the test dataset is treated as a standalone prefix.
> > > We believe that the [Missing Experiments - Varying the Number of Shots] section in our response to Reviewer AoXe could help clarify your understanding on this point.
> > > For ReCaLL, we provide baselines using different compositions of shots (Rand, RandM, and RandNM) to construct a prefix.
> > >
> > > Regarding your point on WikiMIA, please note that we have also evaluated our method using OLMoMIA, further demonstrating its applicability.
> > >
> > > Could you please elaborate on the persisting concerns regarding the novelty of our paper?
> > > Within the literature on membership inference attacks for large language models, we believe our contributions—including the iterative algorithm to refine membership scores and the introduction of new datasets with varying difficulty based on clustering—represent significant novelty.
> > >
> > > If our previous response was helpful and our current response addresses your remaining concerns, we kindly ask you to consider raising your score.

---

### Official Review · Reviewer_AoXe · 2024-11-05

**Soundness:** 2
**Presentation:** 3
**Contribution:** 3
**Rating:** 5
**Confidence:** 4

**Summary:**

The authors propose a membership inference attack for LLMs. Their key insight is that previous work requires non-member prompts for good membership inference, so one can bootstrap good prompts from membership scores and vice versa. This leads them to iteratively improving (non-member prompts) and membership scores. Empirically, the proposed method exhibits significant improvements in the attack AUROC when compared with prior art.

**Strengths:**

* The idea of using prefix scores to refine membership scores and vice versa is novel.
* The empirical results are also quite strong, which leads me to believe that the proposed method will be a good benchmark for future methods.
* I commend the authors for the care they have taken to design a new benchmark. I believe it is quite significant and will be valuable for future work.

**Weaknesses:**

## Main concerns
These major considerations need to be addressed for acceptance, in my opinion:

* **Metric**: The paper reports the AUROC only but not [TPR @ low FPR](https://arxiv.org/pdf/2112.03570), which is the better metric per community consensus. It is fine to design the method based on AUROC but TPR @ low FPR should be reported too.

* **Computational complexity**: * The paper lacks a precise and quantitative description of the computational complexity. I would recommend the authors to describe the complexity of the proposed method (e.g. in terms of tokens consumed by the LLM) and an apples-to-apples comparisons with baselines. Some factors to take into account are the number of iterations, number of shots, cost of computing the prefix/membership score, etc.

* **Missing experiments**: While the results are strong, the coverage/ablations of the current experiments can be greatly improved. Examples:
    - Varying number of shots for the proposed method and ReCaLL. I expect the proposed method to be more robust than ReCaLL, but a plot to this effect is missing.
    - Compare baselines in a compute-constrained setting, so that all methods receive the same computational budget.
    - Line 359 says that "EM-MIA requires a baseline sufficiently better than random guessing". How much better? I would like to see some ablations with different initialization methods to understand how good the initialization must be.
    - Vary the number of iterations of the proposed method.
    - Lines 381-395: the effect of reusing test examples for ReCaLL needs to be explored through ablations too.
These experiments are very valuable for the community. For example, how robust your method is to variations in the hyperparameters?

## Other suggestions/comments

* **Clarity**:
    - It is not clear initially if the paper deals with MIA in the pretraining or finetuning settings. It would be helpful to clarify that upfront.
    - The authors should provide some intuition of *why* ReCaLL works, given that it is a super recent development.
    - "Prefix score" in the abstract is very ambiguous.
    - It is hard to interpret the experimental results as the tables are full of numbers. The authors may wish to present the results for one length and relegate the rest to the appendix. It would also be helpful to use some plots to demonstrate results.

* **OLMoMIA design not clear**: The second half of section 5 is written in a very casual manner and is ambiguous. I do not understand how the easy/medium/hard settings are designed. Further, are members and non-members clustered together or separately (Line 322)? I would recommend the use of more precise language (e.g. with equations). Alternatively, some figures here can greatly help (with pseudocode in the appendix). For instance, Fig 2 is quite nice.

* **Missing refs**: [Kandpal et al. (EMNLP '24)](https://arxiv.org/abs/2310.09266), [Maini et al. (NeurIPS '24)](https://arxiv.org/abs/2406.06443)

**Questions:**

* Why use the ratio of the log-likelihoods instead of the difference? Theory (e.g. Neyman-Pearson lemma) suggests very strongly that the ratio of the likelihoods (i.e. difference of log-likelihoods) is the right operation, and the ratio of the log likelihoods can be degenerate is some circumstances. Can the authors try out a variant of ReCaLL with the difference of log likelihoods?

* The OLMoMIA benchmark only appears to work with the OLMo model. Do we have any evidence that MIA results on one model will transfer to another when factors such as distribution shift are controlled appropriately?

* Why do the authors strongly emphasize expectation maximization? This is super puzzling to me because EM is a very specific optimization algorithm used in the context of latent variable models or missing values by maximizing the ELBO (which is a provable lower bound, as described in Sec. 8.7.2 of [Murphy's book](https://probml.github.io/pml-book/book1.html)). The only resemblance I see is that there is iterative/alternating optimization, but that is very common in machine learning, optimization, statistics, etc. Unless I'm missing some deeper insight, I would suggest that the paper would be better off without a shallow and misleading comparison to EM.

* What are your plans for releasing code/software?

---

> ### Author Response · Authors · 2024-11-26
> **Rebuttal to Reviewer AoXe**
>
> We appreciate you found our work novel, empirically strong, and valuable for future research. We have made every effort to address your concerns and incorporate your suggestions. In some cases, we provided clarifications or requested additional feedback to improve our paper further. We hope you might consider raising your score to an acceptance level if our responses adequately address the major considerations you identified as necessary for acceptance.
>
> ---
>
> ### **[Metric - TPR @ low FPR]**
> Please see General Response (and Appendix B).
>
> ---
>
> ### **[Computational Complexity]**
> Please see General Response (and Appendix C). We agree that considering a compute-constrained setting is a valuable suggestion. We view improving the efficiency of EM-MIA and other MIAs as an exciting direction for future work.
>
> ---
>
> ### **[Missing Experiments - Varying the number of shots $n$]**
> To clarify, while each component in EM-MIA is a design choice, we mainly used the update rule for prefix scores as Line 3 in Algorithm 1 (calculating pairwise ReCaLL scores on all data $x \in D_{test}$ using each example $p \in D_{test}$ as a standalone prefix once and comparing them with the estimate of membership scores at each iteration) and the update rule for membership scores as Line 4 in Algorithm 1 (directly using a negative prefix score as a membership score), as described in Section 6.4. In this case, we do not even need to tune or exhaustively search $n$. That is the reason why we do not include a plot for the effect of $n$.
>
> On the other hand, $n$ is an important hyperparameter for ReCaLL. You can refer to the ReCaLL paper to see how its performance changes with respect to $n$. In our experience, ReCaLL was sensitive to $n$ and the selected data points to construct a prefix. Our Figure 1 illustrates how the effectiveness of each example as a prefix varies significantly.
>
> If you mentioned the number of shots because you are thinking about choosing a prefix consisting of data points with top prefix scores that we suggested as an alternative way to update membership scores, we hope this explanation resolves your concern. We will further clarify this in the revision.
>
> ---
>
> ### **[Missing Experiments - Varying the number of iterations]**
> As explained above, each iteration updates scores without requiring additional LLM inferences, making the process fast. The number of iterations does not affect the overall computational complexity. We set the maximum number to 10, and the process usually converged after 4-5 iterations.
>
> ---
>
> ### **[Missing Experiments - Ablation on initialization]**
> Please see General Response (and Appendix A).
>
> ---
>
> ### **[Missing Experiments - Reusing test examples]**
> Could you please elaborate more on the ablation experiments you described? If you want a comparison after removing data points used from ReCaLL, this can be done easily by omitting the “keep_used” option in “eval.py.”
>
> ---
>
> ### **[Clarity - Pre-training data detection]**
> Our method can indeed be applied to different training stages, but we focused on pre-training data detection in this work. We will clarify this in the revision.
>
> ---
>
> ### **[Clarity - Intuition behind ReCaLL]**
> Thank you for this suggestion. We added a brief explanation to provide an intuition of why ReCaLL works in Section 2.4 for readers unfamiliar with ReCaLL.
>
> ---
>
> ### **[Clarity - Prefix score]**
> We agree that the definition of a prefix score in the abstract might feel abstract. Providing a formal definition would require a detailed explanation of ReCaLL, which could overwhelm readers. If you have suggestions for simplifying this explanation without losing clarity, we would greatly appreciate them.
>
> ---
>
> ### **[Clarity - Table]**
> We recognize the tables are dense with numbers. Following Min-K%++ and ReCaLL, we reported results for all combinations of methods, models, and lengths. Our tables include additional results due to the larger number of ReCaLL-based baselines we evaluated. We encourage readers to first focus on average scores and then dive into specific comparisons of interest.
>
> ---
>
> ### **[Clarity - OLMoMIA]**
> We added equations for our OLMoMIA settings in Appendix E. Members and non-members are clustered separately. We stated this in Line 326 of the original manuscript.
>
> ---
>
> ### **[Missing References]**
> Thank you for pointing out missing references. We have added them in the revised version.
>
> ---
>
> ### **[Why use the ratio of the log-likelihoods instead of the difference?]**
> We experimented with both the ratio and the difference of log-likelihoods. Empirically, the ratio performed slightly better overall. Additionally, we followed the ReCaLL approach for consistency in comparison.
>
> ---
>
> ### **[EM Interpretation]**
> We interpret prefix scores as latent variables that need to be predicted.
>
> ---
>
> ### **[Plans for releasing code]**
> We uploaded our code as supplementary material.

---

> > ### Comment · Reviewer_AoXe · 2024-11-26
> >
> > Thanks for the response. The additional results on TPR at low FPR are well received and it is great that the authors are willing to release code.
> >
> > Follow-up discussions:
> >
> > - Number of shots: are you suggesting that $n=1$ for your method? If yes, this is not clear from reading the paper and the authors may wish to clarify this. If not, this needs a plot.
> > - Computational complexity: please report wall-clock times.
> > - Prefix score: Lines 011-019 of the abstract can possibly be replaced with 2-3 lines of description which explains how the membership score is constructed (using non-member prefixes) and that the prefix score measures how discriminating a prefix is for membership inference.
> > - Table: I strongly suggest better presentation.
> > - Ratio of log likelihoods: Please report the numbers to justify the decision of taking a ratio instead of the difference of log likelihoods. I did not find a justification/comparison in the ReCALL paper, so that does not make "we followed ReCALL" a fully sound design decision.
> > - EM interpretation: Can the authors justify this more rigorously by stating a formal probabilistic model in which the prefix scores are latent variables? If not, then this interpretation is incorrect and misleading. I strongly suggest a change in nomenclature in this case.
> >
> > Overall, I wish to maintain my score as I believe that it is an accurate reflection of the overall rating of the paper.

---

> > > ### Author Response · Authors · 2024-12-03
> > >
> > > Thank you for your thoughtful response and for providing follow-up discussions.
> > >
> > > We sincerely hope our response has adequately addressed your major concerns and that you will consider them positively.
> > >
> > > We will incorporate your suggestions to improve the presentation of our paper, specifically regarding the number of shots, prefix scores, and table formatting. Additionally, we confirm that $n=1$ in our method, and we will ensure this is clarified in the revised version.

---

### Author Response · Authors · 2024-11-26
**General Response**

We sincerely appreciate the time and effort you invested in reading our paper and providing feedback. Your insights have been instrumental in significantly improving the quality of our work. We addressed the major concerns as follows.

---

### **[Ablation with Different Initializations and Scoring Functions]** (AoXe, dgtr, zPaX):
We added an ablation study in Appendix A, where we analyzed the performance of EM-MIA under different combinations of initializations and scoring functions. The results show that EM-MIA consistently converges to a similar AUC-ROC, reinforcing the robustness discussed in Section 6.3.

---

### **[Experimental Results with TPR @ low FPR Metric]** (AoXe, dgtr):
We included TPR@1%FPR results in Appendix B, highlighting the utility of this metric for MIA evaluation, as discussed in Section 2.1. These results further underscore the superiority of EM-MIA compared to other methods.

---

### **[Discussion on Computational Costs]** (AoXe, dgtr, zPaX):
We added a general discussion on computational costs in Section 7.3 and a detailed comparison of computational complexity between EM-MIA and other baselines in Appendix C. While MIA accuracy is typically prioritized over computational efficiency, we found that experiments on the benchmarks used were sufficiently tractable for all methods.

---

Additionally, we improved the overall writing by incorporating clarifications, additional background information, and missing references. We also uploaded our code (AoXe, HkGC) as supplementary material to enhance transparency and support future research. Finally, we provided detailed responses to the remaining comments individually.

---

Thank you again for your thoughtful feedback. It has been invaluable in refining our work.

---

### Meta-Review · Area_Chair_WK3Q · 2024-12-17

**Metareview:**

This paper introduces a new method for Membership Inference Attack (MIA), EM-MIA. EM-MIA is an iterative algorithm based on the expectation-maximization (EM) framework, which jointly refines membership and prefix scores to improve accuracy. The paper also introduces OLMoMIA, a benchmark allowing controlled experiments on MIA difficulty levels by adjusting training and test data distribution overlap. Reviewers had several concerns about the paper, including not reporting TPR @low FPR (which was resolved in the author response) and the limitation of wikiMIA dataset (no experiments on MIMIR dataset). Overall the paper seems to be heading in the right direction but the reviews suggest stronger empirical results would be required.

**Additional Comments On Reviewer Discussion:**

One reviewer stated that a Theorem implies the impossibility of MIA, however that theorem does not seem to apply to this setting. Therefore the score of that reviewer was not taken into account in this meta-review.

---

### Decision · Program_Chairs · 2025-01-22

Reject